# Inductive and Unsupervised Representation Learning on Graph Structured Objects

**Lichen Wang**[1], **Bo Zong**[2], **Qianqian Ma**[3], **Wei Cheng**[2], **Jingchao Ni**[2], **Wenchao Yu**[2],
**Yanchi Liu**[2], **Dongjin Song**[2], **Haifeng Chen**[2], **and Yun Fu**[1]
[1]Northeastern University, Boston, USA
[2]NEC Laboratories America, Princeton, USA
[3]Boston University, Boston, USA
`wanglichenxj@gmail.com, maqq@bu.edu, yunfu@ece.neu.edu,`
`{bzong,weicheng,jni,wyu,yanchi,dsong,heifeng}@nec-labels.com`

## Abstract

Inductive and unsupervised graph learning is a critical technique for predictive or information retrieval tasks where label information is difficult to obtain. It is also challenging to make graph learning inductive and unsupervised at the same time, as learning processes guided by reconstruction error based loss functions inevitably demand graph similarity evaluation that is usually computationally intractable. In this paper, we propose a general framework SEED (Sampling, Encoding, and Embedding Distributions) for inductive and unsupervised representation learning on graph structured objects. Instead of directly dealing with the computational challenges raised by graph similarity evaluation, given an input graph, the SEED framework samples a number of subgraphs whose reconstruction errors could be efficiently evaluated, encodes the subgraph samples into a collection of subgraph vectors, and employs the embedding of the subgraph vector distribution as the output vector representation for the input graph. By theoretical analysis, we demonstrate the close connection between SEED and graph isomorphism. Using public benchmark datasets, our empirical study suggests the proposed SEED framework is able to achieve up to 10% improvement, compared with competitive baseline methods.

## 1 Introduction

Representation learning has been the core problem of machine learning tasks on graphs. Given a graph structured object, the goal is to represent the input graph as a dense low-dimensional vector so that we are able to feed this vector into off-the-shelf machine learning or data management techniques for a wide spectrum of downstream tasks, such as classification (Niepert et al., 2016), anomaly detection (Akoglu et al., 2015), information retrieval (Li et al., 2019), and many others (Santoro et al., 2017b; Nickel et al., 2015).

In this paper, our work focuses on learning graph representations in an inductive and unsupervised manner. As inductive methods provide high efficiency and generalization for making inference over unseen data, they are desired in critical applications. For example, we could train a model that encodes graphs generated from computer program execution traces into vectors so that we can perform malware detection in a vector space. During real-time inference, efficient encoding and the capability of processing unseen programs are expected for practical usage. Meanwhile, for real-life applications where labels are expensive or difficult to obtain, such as anomaly detection (Zong et al., 2018) and information retrieval (Yan et al., 2005), unsupervised methods could provide effective feature representations shared among different tasks.

Inductive and unsupervised graph learning is challenging, even compared with its transductive or supervised counterparts. First, when inductive capability is required, it is inevitable to deal with the problem of node alignment such that we can discover common patterns across graphs. Second, in the case of unsupervised learning, we have limited options to design objectives that guide learning processes. To evaluate the quality of the learned latent representations, reconstruction errors are

Figure 1: SEED consists of three components: sampling, encoding, and embedding distribution. Given an input graph, its vector representation can be obtained by going through the components.

commonly adopted. When node alignment meets reconstruction error, we have to answer a basic question: Given two graphs $\mathcal{G}_1$ and $\mathcal{G}_2$, are they identical or isomorphic (Chartrand, 1977)? To this end, it could be computationally intractable to compute reconstruction errors (*e.g.*, using graph edit distance (Zeng et al., 2009) as the metric) in order to capture detailed structural information.

Previous deep graph learning techniques mainly focus on transductive (Perozzi et al., 2014) or supervised settings (Li et al., 2019). A few recent studies focus on autoencoding specific structures, such as directed acyclic graphs (Zhang et al., 2019), trees or graphs that can be decomposed into trees (Jin et al., 2018), and so on. From the perspective of graph generation, You et al. (2018) propose to generate graphs of similar graph statistics (*e.g.*, degree distribution), and Bojchevski et al. (2018) provide a GAN based method to generate graphs of similar random walks.

In this paper, we propose a general framework SEED (Sampling, Encoding, and Embedding Distributions) for inductive and unsupervised representation learning on graph structured objects. As shown in Figure 1, SEED consists of three major components: subgraph sampling, subgraph encoding, and embedding subgraph distributions. SEED takes arbitrary graphs as input, where nodes and edges could have rich features, or have no features at all. By sequentially going through the three components, SEED outputs a vector representation for an input graph. One can further feed such vector representations to off-the-shelf machine learning or data management tools for downstream learning or retrieval tasks.

Instead of directly addressing the computational challenge raised by evaluation of graph reconstruction errors, SEED decomposes the reconstruction problem into the following two sub-problems.

*Q1: How to efficiently autoencode and compare structural data in an unsupervised fashion?* SEED focuses on a class of subgraphs whose encoding, decoding, and reconstruction errors can be evaluated in polynomial time. In particular, we propose random walks with earliest visiting time (WEAVE) serving as the subgraph class, and utilize deep architectures to efficiently autoencode WEAVEs. Note that reconstruction errors with respect to WEAVEs are evaluated in linear time.

*Q2: How to measure the difference of two graphs in a tractable way?* As one subgraph only covers partial information of an input graph, SEED samples a number of subgraphs to enhance information coverage. With each subgraph encoded as a vector, an input graph is represented by a collection of vectors. If two graphs are similar, their subgraph distribution will also be similar. Based on this intuition, we evaluate graph similarity by computing distribution distance between two collections of vectors. By embedding distribution of subgraph representations, SEED outputs a vector representation for an input graph, where distance between two graphs' vector representations reflects the distance between their subgraph distributions.

Unlike existing message-passing based graph learning techniques whose expressive power is upper bounded by Weisfeiler-Lehman graph kernels (Xu et al., 2019; Shervashidze et al., 2011), we show the direct relationship between SEED and graph isomorphism in Section 3.5.

We empirically evaluate the effectiveness of the SEED framework via classification and clustering tasks on public benchmark datasets. We observe that graph representations generated by SEED are able to effectively capture structural information, and maintain stable performance even when the node attributes are not available. Compared with competitive baseline methods, the proposed SEED framework could achieve up to 10% improvement in prediction accuracy. In addition, SEED

achieves high-quality representations when a reasonable number of small subgraph are sampled. By adjusting sample size, we are able to make trade-off between effectiveness and efficiency.

## 2 RELATED WORK

**Kernel methods**. Similarity evaluation is one of the key operations in graph learning. Conventional graph kernels rely on handcrafted substructures or graph statistics to build vector representations for graphs (Borgwardt & Kriegel, 2005; Kashima et al., 2003; Vishwanathan et al., 2010; Horváth et al., 2004; Shervashidze & Borgwardt, 2009; Kriege et al., 2019). Although kernel methods are potentially unsupervised and inductive, it is difficult to make them handle rich node and edge attributes in many applications, because of the rigid definition of substructures.

**Deep learning**. Deep graph representation learning suggests a promising direction where one can learn unified vector representations for graphs by jointly considering both structural and attribute information. While most of existing works are either transductive (Perozzi et al., 2014; Kipf & Welling, 2016; Liu et al., 2018) or supervised settings (Scarselli et al., 2008; Battaglia et al., 2016; Defferrard et al., 2016; Duvenaud et al., 2015; Kearnes et al., 2016; Veličković et al., 2018; Santoro et al., 2017a; Xu et al., 2018; Hamilton et al., 2017), a few recent studies focus on autoencoding specific structures, such as directed acyclic graphs (Zhang et al., 2019), trees or graphs that can be decomposed into trees (Jin et al., 2018), and so on. In the case of graph generation, You et al. (2018) propose to generate graphs of similar graph statistics (*e.g.*, degree distribution), and Bojchevski et al. (2018) provide a method to generate graphs of similar random walks. In addition, Li et al. (2019) propose a supervised method to learn graph similarity, and Xu et al. (2019) theoretically analyses the expressive power of existing message-passing based graph neural networks. Micali & Zhu (2016) propose anonymous walks for reconstruction tasks. It reconstructs a Markov process from the records collected by limited/partial observations. In an anonymous walk procedure, the states are visited according to the underlying transition probabilities, but no global state names are known. Ivanov & Burnaev (2018) deploy anonymous walks as a crucial strategy for obtaining data-driven and feature-based graph representations. An efficient sampling approach is designed which approximates the distributions for large networks.

Unlike existing kernel or deep learning methods, our SEED framework is unsupervised with inductive capability, and naturally supports complex attributes on nodes and edges. Moreover, it works for arbitrary graphs, and provides graph representations that simultaneously capture both structural and attribute information.

## 3 SEED: SAMPLING, ENCODING, AND EMBEDDING DISTRIBUTIONS

The core idea of SEED is to efficiently encode subgraphs as vectors so that we can utilize subgraph distribution distance to reflect graph similarity. We first give an abstract overview on the SEED framework in Section 3.1, and then discuss concrete implementations for each component in Section 3.2, 3.3, and 3.4, respectively. In Section 3.5, we share the theoretical insights in SEED. For the ease of presentation, we focus on undirected graphs with rich node attributes in the following discussion. With minor modification, our technique can also handle directed graphs with rich node and edge attributes.

### 3.1 OVERVIEW

SEED encodes an arbitrary graph into a vector by the following three major components, as shown in Figure 1.

- **Sampling**. A number of subgraphs are sampled from an input graph in this component. The design goal of this component is to find a class of subgraphs that can be efficiently encoded and decoded so that we are able to evaluate their reconstruction errors in a tractable way.

- **Encoding**. Each sampled subgraph is encoded into a vector in this component. Intuitively, if a subgraph vector representation has good quality, we should be able to reconstruct the original subgraph well based on the vector representation. Therefore, the design goal of this component is to find an autoencoding system that provides such encoding functionality.

- **Embedding distribution**. A collection of subgraph vector representations are aggregated into one vector serving as the input graph's representation. For two graphs, their distance in the output vector space approximates their subgraph distribution distance. The design goal of this component is to find such a aggregation function that preserves a pre-defined distribution distance.

Although there could be many possible implementations for the above three components, we propose a competitive implementation in this paper, and discuss them in details in the rest of this section.

## 3.2 SAMPLING

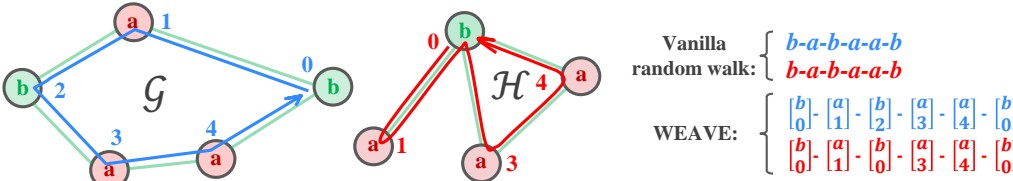

Figure 2: Expressive power comparison between WEAVEs and vanilla random walks: while blue and orange walks cannot be differentiated in terms of vanilla random walks, the difference under WEAVEs is outstanding.

In this paper, we propose to sample a class of subgraphs called WEAVE (random Walk with EArliest Visit timE). Let $\mathcal{G}$ be an input graph of a node set $V(\mathcal{G})$ and an edge set $E(\mathcal{G})$. A WEAVE of length $k$ is sampled from $\mathcal{G}$ as follows.

- **Initialization**. A starting node $v^{(0)}$ is randomly drawn from $V(\mathcal{G})$ at timestamp 0, and its earliest visiting time is set to 0.

- **Next-hop selection**. Without loss of generality, assume $v^{(p)}$ is the node visited at timestamp $p$ ($0 \leq p < k$). We randomly draw a node $v^{(p+1)}$ from $v^{(p)}$'s one-hop neighborhood as the node to be visited at timestamp $p+1$. If $v^{(p+1)}$ is a node that we have not visited before, its earliest visiting time is set to $p+1$; otherwise, its earliest visiting is unchanged. We hop to $v^{(p+1)}$.

- **Termination**. The sampling process ends when timestamp reaches $k$.

In practical computation, a WEAVE is denoted as a matrix $X = [\mathbf{x}^{(0)}, \mathbf{x}^{(1)}, \cdots, \mathbf{x}^{(k)}]$. In particular, $\mathbf{x}^{(p)} = [\mathbf{x}_a^{(p)}, \mathbf{x}_t^{(p)}]$ is a concatenation of two vectors, where $\mathbf{x}_a^{(p)}$ includes attribute information for the node visited at timestamp $p$, and $\mathbf{x}_t^{(p)}$ contains its earliest visit time. As earliest visit time is discrete, we use one-hot scheme to represent such information, where $\mathbf{x}_t^{(p)}$ is a $k$-dimensional vector and $\mathbf{x}_t^{(p)}[q] = 1$ means the earliest visit time is timestamp $q$. If one aims to sample $s$ WEAVEs from an input graph, the output of this component is a set of $s$ matrices $\{X_1, X_2, ..., X_s\}$.

**Difference between WEAVEs and vanilla random walks**. The key distinction comes from the information of the earliest visit time. Vanilla random walks include coarser-granularity structural information, such as neighborhood density and neighborhood attribute distribution (Perozzi et al., 2014). As vanilla random walks have no memory on visit history, detailed structural information related to loops or circles is ignored. While it is also efficient to encode and decode vanilla random walk, it is difficult to evaluate finer-granularity structural difference between graphs. Unlike vanilla random walks, WEAVEs utilize earliest visit time to preserve loop information in sampled subgraphs. As shown in Figure 2, while we cannot tell the difference between walk $w_1$ and walk $w_2$ using vanilla random walk, the distinction is outstanding under WEAVEs. Note that it is equally efficient to encode and decode WEAVEs, compared with vanilla random walks.

In addition, WEAVE is also related to anonymous random walks (Ivanov & Burnaev, 2018; Micali & Zhu, 2016). By excluding attribution information, a WEAVE is reduced to an anonymous random walk.

### 3.3 ENCODING

Given a set of sampled WEAVEs of length $k$ $\{X_1, X_2, ..., X_s\}$, the goal is to encode each sampled WEAVE into a dense low-dimensional vector. As sampled WEAVEs share same length, their matrix representations also have identical shapes. Given a WEAVE $X$, one could encode it by an autoencoder (Hinton & Salakhutdinov, 2006) as follows.

$$\mathbf{z} = f(X; \theta_e), \qquad\qquad \hat{X} = g(\mathbf{z}; \theta_d), \qquad\qquad (1)$$

where $\mathbf{z}$ is the dense low-dimensional representation for the input WEAVE, $f(\cdot)$ is the encoding function implemented by an MLP with parameters $\theta_e$, and $g(\cdot)$ is the decoding function implemented by another MLP with parameters $\theta_d$. The quality of $\mathbf{z}$ is evaluated through reconstruction errors as follows,

$$\mathcal{L} = \|X - \hat{X}\|_2^2. \qquad\qquad (2)$$

By conventional gradient descent based backpropagation (Kingma & Ba, 2014), one could optimize $\theta_e$ and $\theta_d$ via minimizing reconstruction error $\mathcal{L}$. After such an autoencoder is well trained, the latent representation $\mathbf{z}$ includes both node attribute information and finer-granularity structural information simultaneously. Given $s$ sampled WEAVEs of an input graph, the output of this component is $s$ dense low-dimensional vectors $\{\mathbf{z}_1, \mathbf{z}_2, \cdots, \mathbf{z}_s\}$.

### 3.4 EMBEDDING DISTRIBUTION

Let $\mathcal{G}$ and $\mathcal{H}$ be two arbitrary graphs. Suppose subgraph (*e.g.*, WEAVE) distributions for $\mathcal{G}$ and $\mathcal{H}$ are $P_\mathcal{G}$ and $P_\mathcal{H}$, respectively. In this component, we are interested in evaluating the distance between $P_\mathcal{G}$ and $P_\mathcal{H}$. In this work, we investigate the feasibility of employing empirical estimate of the maximum mean discrepancy (MMD) (Gretton et al., 2012) to evaluate subgraph distribution distances, without assumptions on prior distributions, while there are multiple candidate metrics for distribution distance evaluation, such as KL-divergence (Kullback & Leibler, 1951) and Wasserstein distance (Arjovsky et al., 2017). We leave the detailed comparison among different choices of distance metrics in our future work.

Given $s$ subgraphs sampled from $\mathcal{G}$ as $\{\mathbf{z}_1, \cdots, \mathbf{z}_s\}$ and $s$ subgraphs sampled from $\mathcal{H}$ as $\{\mathbf{h}_1, \cdots, \mathbf{h}_s\}$, we can estimate the distance between $P_\mathcal{G}$ and $P_\mathcal{H}$ under the MMD framework:

$$\widehat{MMD}(P_\mathcal{G}, P_\mathcal{H}) = \frac{1}{s(s-1)} \sum_{i=1}^{s} \sum_{j \neq i}^{s} k(\mathbf{z}_i, \mathbf{z}_j) + \frac{1}{s(s-1)} \sum_{i=1}^{s} \sum_{j \neq i}^{s} k(\mathbf{h}_i, \mathbf{h}_j)$$
$$- \frac{2}{s^2} \sum_{i=1}^{s} \sum_{j=1}^{s} k(\mathbf{z}_i, \mathbf{h}_j)$$
$$= \|\hat{\mu}_\mathcal{G} - \hat{\mu}_\mathcal{H}\|_2^2. \qquad\qquad (3)$$

$\hat{\mu}_\mathcal{G}$ and $\hat{\mu}_\mathcal{H}$ are empirical kernel embeddings of $P_\mathcal{G}$ and $P_\mathcal{H}$, respectively, and are defined as follows,

$$\hat{\mu}_\mathcal{G} = \frac{1}{s} \sum_{i=1}^{s} \phi(\mathbf{z}_i), \qquad\qquad \hat{\mu}_\mathcal{H} = \frac{1}{s} \sum_{i=1}^{s} \phi(\mathbf{h}_i), \qquad\qquad (4)$$

where $\phi(\cdot)$ is the implicit feature mapping function with respect to the kernel function $k(\cdot, \cdot)$. To this end, $\hat{\mu}_\mathcal{G}$ and $\hat{\mu}_\mathcal{H}$ are the output vector representation for $\mathcal{G}$ and $\mathcal{H}$, respectively.

In terms of kernel selection, we find the following options are effective in practice.

**Identity kernel**. Under this kernel, pairwise similarity evaluation is performed in the original input space. Its implementation is simple, but surprisingly effective in real-life datasets,

$$\hat{\mu}_\mathcal{G} = \frac{1}{s} \sum_{i=1}^{s} \mathbf{z}_i, \qquad\qquad \hat{\mu}_\mathcal{H} = \frac{1}{s} \sum_{i=1}^{s} \mathbf{h}_i. \qquad\qquad (5)$$

where output representations are obtained by average aggregation over subgraph representations.

**Commonly adopted kernels**. For popular kernels (*e.g.*, RBF kernel, inverse multi-quadratics kernel, and so on), it could be difficult to find and adopt their feature mapping functions. While approximation methods could be developed for individual kernels (Ring & Eskofier, 2016), we could train

a deep neural network that approximates such feature mapping functions. In particular,

$$\hat{\mu}'_{\mathcal{G}} = \frac{1}{s} \sum_{i=1}^{s} \hat{\phi}(\mathbf{z}_i; \theta_m), \qquad \hat{\mu}'_{\mathcal{H}} = \frac{1}{s} \sum_{i=1}^{s} \hat{\phi}(\mathbf{h}_i; \theta_m), \qquad D(P_{\mathcal{G}}, P_{\mathcal{H}}) = \|\hat{\mu}'_{\mathcal{G}} - \hat{\mu}'_{\mathcal{H}}\|_2^2, \quad (6)$$

where $\hat{\phi}(\cdot; \theta_m)$ is an MLP with parameters $\theta_m$, and $D(\cdot, \cdot)$ is the approximation to the empirical estimate of MMD. Note that $\hat{\mu}'_{\mathcal{G}}$ and $\hat{\mu}'_{\mathcal{H}}$ are output representations for $\mathcal{G}$ and $\mathcal{H}$, respectively. To train the function $\hat{\phi}(\cdot; \theta_m)$, we evaluate the approximation error by

$$J(\theta_m) = \|D(P_{\mathcal{G}}, P_{\mathcal{H}}) - \widehat{MMD}(P_{\mathcal{G}}, P_{\mathcal{H}})\|_2^2, \qquad (7)$$

where $\theta_m$ is optimized by minimizing $J(\theta_m)$.

## 3.5 THEORETICAL INSIGHTS

In this section, we sketch the theoretical connection between SEED and well-known graph isomorphism (Chartrand, 1977), and show how walk length in WEAVE impacts the effectiveness in graph isomorphism tests. The full proof of theorems and lemmas is detailed in Appendix.

To make the discussion self-contained, we define graph isomorphism and its variant with node attributes as follows.

**Graph isomorphism**. $\mathcal{G} = (V(\mathcal{G}), E(\mathcal{G}))$ and $\mathcal{H} = (V(\mathcal{H}), E(\mathcal{H})))$ are isomorphic if there is a bijection function $f : V(\mathcal{G}) \Leftrightarrow V(\mathcal{H})$ such that $\forall (u, v) \in E(\mathcal{G}) \Leftrightarrow (f(u), f(v)) \in E(\mathcal{H})$.

**Graph isomorphism with node attributes**. Let $\mathcal{G} = (V(\mathcal{G}), E(\mathcal{G}), l_1)$, $\mathcal{H} = (V(\mathcal{H}), E(\mathcal{H}), l_2)$ be two attributed graphs, where $l_1, l_2$ are attribute mapping functions $l_1 : V(\mathcal{G}) \to \mathbb{R}^d$, $l_2 : V(\mathcal{H}) \to \mathbb{R}^d$, and node attributes are denoted as $d$-dimensional vectors. Then $\mathcal{G}$ and $\mathcal{H}$ are isomorphic with node attributes if there is a bijection $f : V(\mathcal{G}) \Leftrightarrow V(\mathcal{H})$, s.t., $\forall (u, v) \in E(\mathcal{G}) \Leftrightarrow (f(u), f(v)) \in E(\mathcal{H})$, and $\forall u \in V(\mathcal{G}), l_1(u) = l_2(f(u))$.

**Identical distributions**. Two distributions $P$ and $Q$ are identical if and only if their 1st order Wasserstein distance (Rüschendorf, 1985) $W_1(P, Q) = 0$.

The following theory suggests the minimum walk length for WEAVEs, if every edge in a graph is expected to be visited.

**Lemma 1.** *Let $\mathcal{G} = (V(\mathcal{G}), E(\mathcal{G}))$ be a connected graph, then there exists a walk of length $k$ which can visit all the edges of $\mathcal{G}$, where $k \geq 2|E(\mathcal{G})| - 1$.*

Now, we are ready to present the connection between SEED and graph isomorphism.

**Theorem 1.** *Let $\mathcal{G} = (V(\mathcal{G}), E(\mathcal{G}))$ and $\mathcal{H} = (V(\mathcal{H}), E(\mathcal{H}))$ be two connected graphs. Suppose we can enumerate all possible WEAVEs from $\mathcal{G}$ and $\mathcal{H}$ with a fixed-length $k \geq 2 \max\{|E(\mathcal{G})|, |E(\mathcal{H})|\} - 1$, where each WEAVE has a unique vector representation generated from a well-trained autoencoder. The Wasserstein distance between $\mathcal{G}$'s and $\mathcal{H}$'s WEAVE distributions is $0$ if and only if $\mathcal{G}$ and $\mathcal{H}$ are isomorphic.*

The following theory shows the connection in the case of graphs with nodes attributes.

**Theorem 2.** *Let $\mathcal{G} = (V(\mathcal{G}), E(\mathcal{G}))$ and $\mathcal{H} = (V(\mathcal{H}), E(\mathcal{H}))$ be two connected graphs with node attributes. Suppose we can enumerate all possible WEAVEs on $\mathcal{G}$ and $\mathcal{H}$ with a fixed-length $k \geq 2 \max\{|E(\mathcal{G})|, |E(\mathcal{H})|\} - 1$, where each WEAVE has a unique vector representation generated from a well-trained autoencoder. The Wasserstein distance between $\mathcal{G}$'s and $\mathcal{H}$'s WEAVE distributions is $0$ if and only if $\mathcal{G}$ and $\mathcal{H}$ are isomorphic with node attributes.*

Note that similar results can be easily extended to the cases with both node and edge attributes, the corresponding details can be found in Appendix E.

The theoretical results suggest the potential power of the SEED framework in capturing structural difference of graph data. As shown above, in order to achieve the same expressive power of graph isomorphism, we need to sample a large number of WEAVEs with a long walk length so that all possible WEAVEs can be enumerated. The resource demand is impractical. However, in the empirical study in Section 4, we show that SEED can achieve state-of-the-art performance, when we sample a small number of WEAVEs with a reasonably short walk length.

## 4 EXPERIMENTS

### 4.1 DATASETS

We employ seven public benchmark datasets to evaluate the effectiveness of SEED. The brief introductions of the datasets are listed below.

- **Deezer User-User Friendship Networks (Deezer)** (Rozemberczki et al., 2018) is a social network dataset which is collected from the music streaming service Deezer. It represents friendship network of users from three European countries (i.e., Romania, Croatia and Hungary). There are three graphs which corresponds to the three countries. Nodes represent the users and edges are the mutual friendships. For the three graphs, the numbers of nodes are $41,773$, $54,573$, and $47,538$, respectively, and the number of edges are $125,826$, $498,202$, and $222,887$, respectively. There exist $84$ distinct genres, and genre notations are considered as node features. Thus, node features are represented as a $84$-dimensional multi-hot vectors.

- **Mutagenic Aromatic and Heteroaromatic Nitro Compounds (MUTAG)** (Debnath et al., 1991) is a chemical bioinformatics dataset, which contains $188$ chemical compounds. The compounds can be divided into two classes according to their mutagenic effect on a bacterium. The chemical data can be converted to graph structures, where each node represents an atom. Explicit hydrogen atoms have been removed. In the obtained graph, the node attributes represent the atom types (i.e., C, N, O, F, I, Cl and Br) while the edge attributes represent bond types (i.e., single, double, triple or aromatic).

- **NCI1** (Wale et al., 2008) represents a balanced subsets of datasets of chemical compounds screened for activity against non-small cell lung cancer and ovarian cancer cell lines, respectively. The label is assigned based on this characteristic. Each compound is converted to a graph. There are $4,110$ graphs in total with $122,747$ edges.

- **PROTEINS** (Borgwardt et al., 2005) is a bioinformatics dataset. The proteins in the dataset are converted to graphs based on the sub-structures and physical connections of the proteins. Specifically, nodes are secondary structure elements (SSEs), and edges represent the amino-acid sequence between the two neighbors. PROTEINS has 3 discrete labels (*i.e.*, *helix*, *sheet*, and *turn*). There are $1,113$ graphs in total with $43,471$ edges.

- **COLLAB** (Leskovec et al., 2005) is a scientific collaboration dataset. It belongs to a social connection network in general. COLLAB is collected from 3 public collaboration datasets (*i.e.*, Astro Physics, Condensed Matter Physics, and High Energy Physics). The ego-networks are generated for individual researchers. The label of each graph represents the field which this researcher belongs to. There are $5,000$ graphs with $24,574,995$ edges.

- **IMDB-BINARY** (Yanardag & Vishwanathan, 2015) is a collaboration dataset of film industry. The ego-network of each actor/actress is converted to a graph data. Each node represents an actor/actress and each edge is the indication if two actors/actresses if they appear in the same movie. IMDB-BINARY has $1,000$ graphs associated with $19,773$ edges in total.

- **IMDB-MULTI** extends the IMDB-BINARY dataset to a multi-class version. It contains a balanced set of ego-networks derived from *Sci-Fi*, *Romance*, and *Comedy* genres. Specifically, there are $1,500$ graphs with $19,502$ edges in total.

### 4.2 BASELINES

Three state-of-the-art representative techniques are implemented as baselines in the experiments.

- **Graph Sample and Aggregate (GraphSAGE)** (Hamilton et al., 2017) is an inductive graph representation learning approach in either supervised or unsupervised manner. GraphSAGE explores node and structure information by sampling and aggregating features from the local neighborhood of each node. A forward propagation algorithm is specifically designed to aggregates the information together. We evaluate GraphSAGE in its unsupervised setting.

- **Graph Matching Network (GMN)** (Li et al., 2019) utilizes graph neural networks to obtain graph representations for graph matching applications. A novel Graph Embedding Network is designed for better preserving node features and graph structures. In particular, Graph Matching

Network is proposed to directly obtain the similarity score of each pair of graphs. In our implementation, we utilize the Graph Embedding Networks and deploy the graph-based loss function proposed in (Hamilton et al., 2017) for unsupervised learning fashion.

- **Graph Isomorphism Network (GIN)** (Xu et al., 2019) provides a simple yet effective neural network architecture for graph representation learning. It deploys the sum aggregator to achieve more comprehensive representations. The original GIN is a supervised learning method. Thus, we follow the GraphSAGE approach, and modify its objective to fit an unsupervised setting.

| Setting | Datasets | Methods Metric | SAGE | GIN | GMN | SEED | SAGE | GIN | GMN | SEED |
|---|---|---|---|---|---|---|---|---|---|---|
| | | | Node Feature **Excluded** | | | | Node Feature **Included** | | | |
| Clustering | Dezzer | ACC | 0.3853 | 0.4913 | 0.4924 | **0.4927** | 0.3840 | **0.4930** | 0.4808 | 0.4810 |
| | | NMI | 0.0079 | 0.0958 | 0.0726 | **0.1277** | 0.0003 | **0.0893** | 0.0651 | 0.0566 |
| | MUTAG | ACC | 0.6649 | 0.4997 | 0.4990 | **0.8014** | 0.6649 | 0.4963 | 0.4910 | **0.7260** |
| | | NMI | 0.0150 | 0.0946 | 0.0825 | **0.3214** | 0.0070 | 0.0933 | 0.0917 | **0.1567** |
| | NCI1 | ACC | 0.5098 | 0.5221 | 0.5022 | **0.5510** | 0.5070 | 0.5204 | 0.5005 | **0.5441** |
| | | NMI | 0.0003 | 0.0015 | 0.0034 | **0.0073** | 0.0002 | 0.0013 | 0.0042 | **0.0089** |
| | PROTEINS | ACC | 0.5657 | 0.5957 | **0.5966** | 0.5957 | 0.5657 | 0.5957 | 0.5957 | **0.5957** |
| | | NMI | 0.0013 | 0.0038 | 0.0117 | **0.0518** | 0.0004 | 0.0034 | 0.0067 | **0.0689** |
| | COLLAB | ACC | 0.5208 | 0.5458 | 0.5173 | **0.5973** | - | - | - | - |
| | | NMI | 0.0025 | 0.0729 | 0.0193 | **0.2108** | - | - | - | - |
| | IMDB-BINARY | ACC | 0.5069 | 0.6202 | 0.5010 | **0.5776** | - | - | - | - |
| | | NMI | 0.0002 | 0.0459 | 0.0093 | **0.0241** | - | - | - | - |
| | IMDB-MULTI | ACC | 0.3550 | 3607 | 0.3348 | **0.3816** | - | - | - | - |
| | | NMI | 0.0019 | 0.0185 | 0.0112 | **0.0214** | - | - | - | - |
| Classification | Dezzer | ACC | 0.3775 | 0.5094 | 0.5427 | **0.6327** | 0.3754 | 0.5270 | 0.5627 | **0.7451** |
| | MUTAG | ACC | 0.6778 | 0.6778 | 0.6889 | **0.8112** | 0.6889 | 0.6778 | 0.6889 | **0.8222** |
| | NCI1 | ACC | 0.5410 | 0.5571 | 0.5123 | **0.6105** | 0.5328 | 0.5231 | 0.5133 | **0.6151** |
| | PROTEINS | ACC | 0.6846 | **0.7387** | 0.6216 | 0.7207 | 0.7027 | 0.7207 | 0.6357 | **0.7462** |
| | COLLAB | ACC | 0.5650 | 0.6170 | 0.5460 | **0.6720** | - | - | - | - |
| | IMDB-BINARY | ACC | 0.5400 | 0.7310 | 0.5140 | **0.7660** | - | - | - | - |
| | IMDB-MULTI | ACC | 0.3866 | 0.3843 | 0.3478 | **0.4466** | - | - | - | - |

Table 1: Evaluating graph representation quality by classification and clustering tasks

Two downstream tasks, classification and clustering, are deployed to evaluate the quality of the learned graph representations.

For classification task, a simple multi-layer fully connected neural network is built as a classifier. We report the average accuracy (ACC) for classification performance. For clustering task, an effective conventional clustering approach, Normalized Cuts (NCut) (Jianbo Shi & Malik, 2000), is used to cluster graph representations. We consider two widely used metrics for clustering performance, including Accuracy (ACC) and Normalized Mutual Information (NMI) (Wu et al., 2009). ACC comes from classification with the best mapping, and NMI evaluates the mutual information across the ground truth and the recovered cluster labels based on a normalization operation. Both ACC and NMI are positive measurements (*i.e.*, the higher the metric is, the better the performance will be).

### 4.3 Performance Analysis

In this section, we discuss the performance of SEED and its baselines in the downstream tasks. The performance with and without the node features are reported. In this set of experiments, SEED adopts identity kernel in the component of embedding distributions.

As shown in Table 1, SEED consistently outperforms the baseline methods in both classification and clustering tasks. For GIN and GMN, supervision information could be crucial in order to differentiate structural variations. As GraphSAGE mainly focuses on aggregating feature information from neighbor nodes, it could be difficult for GraphSAGE to extract effective structural information from an unsupervised manner. In the unsupervised setting, SEED is able to differentiate structural difference at finer granularity and capture rich attribute information, leading to high-quality graph representations with superior performance in downstream tasks. Interestingly, for NCI and PROTEINS datasets, we see node features bring little improvement in the unsupervised setting. One

possible reason could be node feature information has high correlation with structural information in these cases.

| Sampling Number | Classification Accuracy | Clustering ACC | NMI |
|---|---|---|---|
| 25 | 0.6832 | 0.6649 | 0.0031 |
| 50 | 0.6778 | 0.6649 | 0.0005 |
| 100 | 0.7778 | 0.6649 | 0.0537 |
| 150 | 0.7889 | 0.6968 | 0.1081 |
| 200 | 0.7778 | 0.7633 | **0.2100** |
| 300 | 0.7833 | 0.7502 | 0.1995 |
| 400 | **0.8389** | 0.7628 | 0.1928 |
| 800 | 0.8111 | **0.7660** | 0.1940 |

Table 2: Representation quality with different sampling numbers

| Walk Length | Classification Accuracy | Clustering ACC | NMI |
|---|---|---|---|
| 5 | 0.7278 | 0.6649 | 0.0534 |
| 10 | 0.7778 | 0.7633 | 0.2100 |
| 15 | 0.8167 | 0.7723 | 0.2495 |
| 20 | 0.8778 | 0.8245 | 0.3351 |
| 25 | 0.8722 | 0.8218 | **0.3380** |
| 30 | **0.8743** | **0.8285** | 0.3321 |

Table 3: Representation quality with different walk lengths

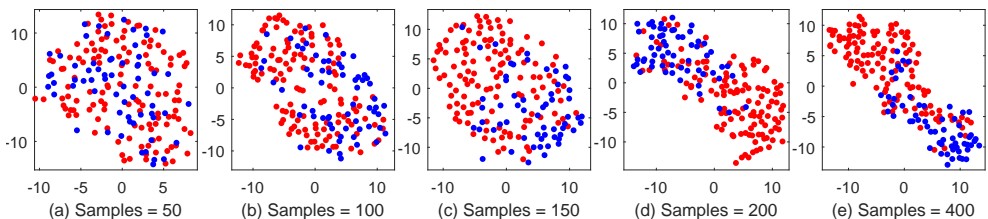

Figure 3: t-SNE visualziation of the MUTAG representations with different sampling numbers

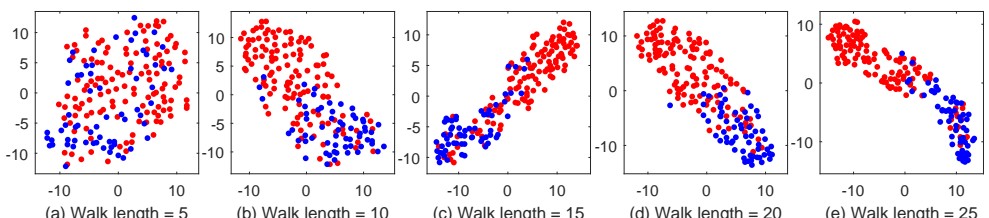

Figure 4: t-SNE visualziation of MUTAG representations with different walk lengths

### 4.4 ABLATION STUDY

Walk length and sample numbers are two meta-parameters in the SEED framework. By adjusting these two meta-parameters, we can make trade-off between effectiveness and computational efficiency. In the experiment, we empirically evaluate the impact of the two meta-parameters on the MUTAG dataset. In Table 2, each row denotes the performance with different sampling numbers (from 25 to 800) while the walk length is fixed to 10. Moreover, we adjust the walk length from 5 to 25 while sampling number is fixed to 200 in Table 3. We can see that the performance of SEED in both classification and clustering tasks increases as there are more subgraphs sampled, especially for the changes from 25 to 200. Meanwhile, we observe the increasing rates diminish dramatically when sampling number ranges from 200 to 800. Similarly, the performance of SEED increase as the walk length grows from 5 to 20, and the performance starts to converge when the length goes beyond 20.

More empirical results are provided in Appendix. In particular, we demonstrate the effectiveness of using DeepSets (Zaheer et al., 2017) for embedding distribution in Appendix F. In Appendix G, we consider a pure feature based random walk baseline with earliest visit time removed, and evaluate its performance. Moreover, we present how to leverage Nyström approximation in embedding distribution, and its performance is reported in Appendix H.

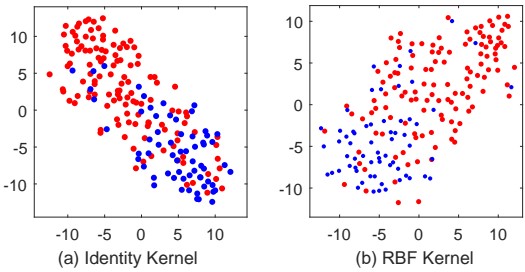

Figure 5: t-SNE visualization of the learned representations from different kernels on MUTAG

## 5 EMBEDDING DISTRIBUTION

We employ t-SNE (Maaten & Hinton, 2008) to visualize learned graph representations in Figure 3 and Figure 4. Red and blue colors indicate two labels. We observe that the boundary becomes clearer when sample number or walk length increases.

| Embedding | Classification ACC | Clustering ACC | Clustering NMI |
|---|---|---|---|
| Identity kernel | 0.8112 | 0.8014 | 0.3214 |
| RBF kernel | 0.7958 | 0.7984 | 0.3115 |

Table 4: Graph representation quality comparison between identity and RBF kernel on MUTAG

Identity kernels or commonly adopted kernels could be deployed in the component of embedding subgraph distributions. In our implementation, we utilize a multi-layer deep neural network to approximate a feature mapping function, for kernels whose feature mapping function is difficult to obtain. Figure 5 shows the t-SNE visualization of learned graph representations based on identity kernel and RBF kernel. As shown in Table 4, SEED variants with different kernels for distribution embedding could distinguish different classes with similar performance on the MUTAG dataset.

## 6 CONCLUSION

In this paper, we propose a novel framework SEED (Sampling, Encoding, and Embedding distribution) framework for unsupervised and inductive graph learning. Instead of directly dealing with the computational challenges raised by graph similarity evaluation, given an input graph, the SEED framework samples a number of subgraphs whose reconstruction errors could be efficiently evaluated, encodes the subgraph samples into a collection of subgraph vectors, and employs the embedding of the subgraph vector distribution as the output vector representation for the input graph. By theoretical analysis, we demonstrate the close connection between SEED and graph isomorphism. Our experimental results suggest the SEED framework is effective, and achieves state-of-the-art predictive performance on public benchmark datasets.

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

## A    PROOF FOR LEMMA 1

*Proof.* We will use induction on $|E(\mathcal{G})|$ to complete the proof.

Basic case: Let $|E(\mathcal{G})| = 1$, the only possible graph is a line graph of length 1. For such a graph, the walk from one node to another can cover the only edge on the graph, which has length $1 = 2 \cdot 1 - 1$.

Induction: We assume for all the connected graphs on less than $m$ edges (*i.e.*, $|E(\mathcal{G})| \leq m - 1$), there exist a walk of length $k$ which can visit all the edges if $k \geq 2|E(\mathcal{G})| - 1$. Then we will show for any connected graph with $m$ edges, there also exists a walk which can cover all the edges on the graph with length $k \geq 2|E(\mathcal{G})| - 1$.

Let $\mathcal{G} = (V(\mathcal{G}), E(\mathcal{G}))$ be a connected graph with $|E(\mathcal{G})| = m$. Firstly, we assume $\mathcal{G}$ is not a tree, which means there exist a cycle on $\mathcal{G}$. By removing an edge $e = (v_i, v_j)$ from the cycle, we can get a graph $\mathcal{G}'$ on $m - 1$ edges which is still connected. This is because any edge on a cycle is not bridge. Then according to the induction hypothesis, there exists a walk $w' = v_1 v_2 \ldots v_i \ldots v_j \ldots v_t$ of length $k' \geq 2(m - 1) + 1$ which can visit all the edges on $\mathcal{G}'$ (The walk does not necessarily start from node 1, $v_1$ just represents the first node appears in this walk). Next, we will go back to our graph $\mathcal{G}$, as $\mathcal{G}'$ is a subgraph of $\mathcal{G}$, $w'$ is also a walk on $\mathcal{G}$. By replacing the first appeared node $v_i$ on walk $w'$ with a walk $v_i v_j v_i$, we can obtain a new walk $w = v_1 v_2 \ldots v_i v_j v_i \ldots v_j \ldots v_t$ on $\mathcal{G}$. As $w$ can cover all the edges on $\mathcal{G}'$ and the edge $e$ with length $k = k' + 2 \geq 2(m - 1) - 1 + 2 = 2m - 1$, which means it can cover all the edges on $\mathcal{G}$ with length $k \geq 2|E(\mathcal{G})| - 1$.

Next, consider graph $\mathcal{G}$ which is a tree. In this case, we can remove a leaf $v_j$ and its incident edge $e = (v_i, v_j)$ from $\mathcal{G}$, then we can also obtain a connected graph $\mathcal{G}'$ with $|E(\mathcal{G}')| = m - 1$. Similarly, according to the induction hypothesis, we can find a walk $w' = v_1 v_2 \ldots v_i \ldots v_t$ on $\mathcal{G}'$ which can visit all the $m - 1$ edges of $\mathcal{G}'$ of length $k'$, where $k' \geq 2(m - 1) - 1$. As $\mathcal{G}'$ is a subgraph of $\mathcal{G}$, any walk on $\mathcal{G}'$ is also a walk on $\mathcal{G}$ including walk $w'$. Then we can also extend walk $w'$ on $\mathcal{G}$ by replacing the first appeared $v_i$ with a walk $v_i v_j v_i$, which produce a new walk $w = v_1 v_2 \ldots v_i v_j v_i \ldots v_t$. $w$ can visit all the edges of $\mathcal{G}'$ as well as the edge $e$ with length $k = k' + 2 \geq 2(m - 1) - 1 + 2 = 2m - 1$. In other words, $w$ can visit all the edges on $\mathcal{G}$ with length $k \geq 2|E(\mathcal{G})| - 1$. Now, we have verified our assumption works for all the connected graphs with $m$ edges, hence we complete our proof. (To give an intuition for our proof of lemma 1, we provide an example of 5 edges in Figure 6)

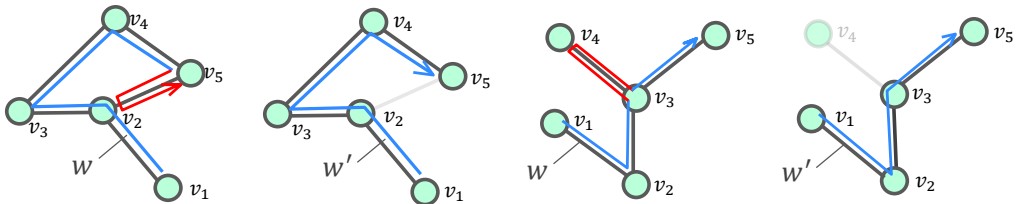

(a1) Graph $\mathcal{G}$ which is not a tree    (a2) Graph $\mathcal{G}'$ corresponds to $\mathcal{G}$ in (a1)    (b1) Graph $\mathcal{G}$ which is a tree    (b2) Graph $\mathcal{G}'$ corresponds to $\mathcal{G}$ in (b1)

Figure 6: Different types of graphs with random walk $w$ which can visit all the edges.

Figure 6 ($a1$) illustrates an example graph $\mathcal{G}$ which is a connected graph on 5 edges but not a tree. By removing an edge $(v_2, v_5)$ from the cycle, we can get a connected graph $\mathcal{G}'$ (Figure 6 ($a2$)) with 4 edges. $\mathcal{G}'$ has a walk $w' = v_1 v_2 v_3 v_4 v_5$ which covers all the edges of $\mathcal{G}'$, as $w'$ is also a walk on $\mathcal{G}$, by replacing $v_5$ with walk $v_5 v_2 v_5$ in $w'$, we can get $w = v_1 v_2 v_3 v_4 v_5 v_2 v_5$ which can visit all the edges of $\mathcal{G}$. Figure 6 ($b1$) shows an example graph $\mathcal{G}$ which is a tree on 5 edges. By removing the leaf $v_4$ and its incident edge $(v_4, v_3)$, we can get a tree $\mathcal{G}'$ with 4 edges (Figure 6 (b2)). $\mathcal{G}'$ has a walk $w' = v_1 v_2 v_3 v_5$ which covers all the edges of $\mathcal{G}'$, as $w'$ is also a walk on $\mathcal{G}$, by replacing $v_3$ with $v_3 v_4 v_3$ in $w'$ we can get a walk $w = v_1 v_2 v_3 v_4 v_3 v_5$ which can cover all the edges of $\mathcal{G}$. ∎

## B    LEMMA 2

The following lemma is crucial for the proof of Theorem 1.

**Lemma 2.** *Suppose that* $w$, $w'$ *are two random walks on graph* $\mathcal{G}$ *and graph* $\mathcal{H}$ *respectively, if the representation of* $w$ *and* $w'$ *are the same, i.e.,* $r_w = r_{w'}$, *the number of the distinct edges on* $w$ *and* $w'$ *are the same, as well as the number of the distinct nodes on* $w$ *and* $w'$.

*Proof.* Let $n_1$, $n_2$ be the number of distinct nodes on $w$, $w'$ respectively, let $m_1$, $m_2$ be the number of distinct edges on $w$ and $w'$ respectively. First, let's prove $n_1 = n_2$. We will prove this by contradiction. Assume $n_1 \neq n_2$, without loss of generality, let $n_1 > n_2$. According to our encoding rule, the largest number appears in a representation vector is the number of the distinct nodes in the corresponding walk. Hence, the largest element in vector $r_w$ is $n_1$ while the largest element in vector $r_{w'}$ is $n_2$. Thus, $r_w \neq r_{w'}$, which contradicts our assumption. Therefore, we have $n_1 = n_2$.

Next, we will show $m_1 = m_2$. We will also prove this point by contradiction. Assume $m_1 \neq m_2$, without loss of generality, let $m_1 > m_2$. As we have proved $n_1 = n_2$, each edge on $w$ and $w'$ will be encoded as a vector like $[k_1, k_2]^\top$, where $k_1, k_2 \in [n_1]$. A walk consists of edges, hence the representation of a walk is formed by the representation of edges. Since $m_1 > m_2$, which means there exists at least two consecutive element $[k_1, k_2]^\top$ in $r_w$ which will not appear in $r_{w'}$, thus $r_w \neq r_{w'}$, which is a contradiction of our assumption. As a result, we can prove $m_1 = m_2$. ∎

## C   PROOF FOR THEOREM 1

*Proof.* We will first prove the sufficiency of the theorem, *i.e.*, suppose graphs $\mathcal{G} = (V(\mathcal{G}), E(\mathcal{G}))$ and $\mathcal{H} = (V(\mathcal{H}), E(\mathcal{H}))$ are two isomorphic graphs, we will show that the WEAVE's distribution on $\mathcal{G}$ and $\mathcal{H}$ are the same.

Let $A$ be the set of all the possible walks with length $k$ on $\mathcal{G}$, $B$ be the set of all the possible walks with length $k$ on $\mathcal{H}$. Each element of $A$ and $B$ represents one unique walk on $\mathcal{G}$ and $\mathcal{H}$ respectively. As we have assumed a WEAVE is a class of subgraphs, which means a WEAVE may corresponds to multiple unique walks in $A$ or $B$. Consider a walk $w = v_1 v_2 \ldots v_i \ldots v_t \in A$ ($v_i$ represent the $i$th node appears in the walk), for any edge $e = (v_i, v_j)$ on $w_i$, as $e \in E(\mathcal{G})$, according to the definition of isomorphism, there exists a mapping $f : V(\mathcal{G}) \to V(\mathcal{H})$ such that $(f(v_i), f(v_j)) \in E(\mathcal{H})$. If we map each node on $w_i$ to graph $\mathcal{H}$, we can get a new walk $w'_i = f(v_1) f(v_2) \ldots f(v_t)$ on $\mathcal{H}$ as each edge $(f(v_i), f(v_j)) \in E(\mathcal{H})$, besides, as the length of $w'_i$ is also $k$, we have $w'_i \in B$. Hence, we can define a new mapping $g : A \to B$, s.t.

$$\forall w_i = v_1 v_2 \ldots v_t \in A, \quad g(w_i) = f(v_1) f(v_2) \ldots f(v_t) = w'_i \in B. \tag{8}$$

Next, we will show that $g$ is a bijective mapping. Firstly, we will show that $f$ is injective. Suppose $g(w_1) = g(w_2)$, we want to show $w_1 = w_2$. Assume $w_1 \neq w_2$, there must exists one step $i$ such that $w_1(i) \neq w_2(i)$, let $w_1(i) = (v_i^{(1)}, v_j^{(1)})$, $w_2(i) = (v_i^{(2)}, v_j^{(2)})$, then we have $(f(v_i^{(1)}), f(v_j^{(1)})) \neq (f(v_i^{(2)}), f(v_j^{(2)}))$ due to the definition of isomorphism. According to the mapping rule of $f$, $(f(v_i^{(1)}), f(v_j^{(1)}))$ is the $i$th step of $f(w_1)$, $(f(v_i^{(2)}), f(v_j^{(2)}))$ is the $i$th step of $g(w_2)$, thus the walk $g(w_1) \neq g(w_2)$, which contradicts our assumption. Therefore, the assumption is false, we have $w_1 = w_2$. Then we will show that $g$ is surjective, *i.e.*, for any $w' \in B$, there exists a $w \in A$ such that $g(w) = w'$. We will also prove this by contradiction, suppose there exists a walk $w' \in B$ such that we can't find any $w \in A$ to make $g(w) = w'$. Let $w' = v_1 v_2 \ldots v_t$, according to the definition of isomorphism, for any edge $(v_i, v_j) \in E(\mathcal{H})$ on $w'$, we have $(f^{-1}(v_i), f^{-1}(v_j)) \in E(\mathcal{G})$, where $f^{-1}$ represents the inverse mapping of $f$. Hence

$$w = f^{-1}(v_1) f^{-1}(v_1) \ldots f^{-1}(v_t) \in A, \tag{9}$$

as $w$ is a walk on graph $\mathcal{H}$ with length $k$. Now consider $g(w)$, based on the mapping rule of $g$, we need to map each node on $w$ via $f$, *i.e.*,

$$g(w) = f(f^{-1}(v_1)) f(f^{-1}(v_1)) \ldots f(f^{-1}(v_t)) = v_1 v_2 \ldots v_t = w', \tag{10}$$

which is contradiction to our assumption. Thus we have proved $g$ is an injective mapping as well as a surjective mapping, then we can conclude that $g$ is a bijective mapping.

Then we will show the WEAVEs' distribution of $\mathcal{G}$ and $\mathcal{H}$ are the same. Since in our assumption, $|E(\mathcal{G})|$ is limited, then $|A|$ and $|B|$ are limited, besides, according to our encoding rule, different

walks may correspond to one specific WEAVE while each WEAVE corresponds a unique representation vector, thus the number of all the possible representation vectors is limited for both $\mathcal{G}$ and $\mathcal{H}$. Thus, the representation vector's distributions $P_{\mathcal{G}}$ for graph $\mathcal{G}$ and representation's distributions $P_{\mathcal{H}}$ for graph $\mathcal{H}$ are both discrete distributions. To compare the similarity of two discrete probability distributions, we can adopt the following equation to compute the Wasserstein distance and check if it is 0.

$$
\begin{aligned}
W_1(\mathbb{P}, \mathbb{Q}) = \min_{\pi} & \sum_{i=1}^{m} \sum_{j=1}^{n} \pi(i,j) s(i,j), \\
s.t. & \sum_{i=1}^{m} \pi(i,j) = w_{q_j}, \forall j, \\
& \sum_{j=1}^{n} \pi(i,j) = w_{p_i}, \forall i, \\
& \pi(i,j) \geq 0, \forall i, j,
\end{aligned}
\tag{11}
$$

where $W_1(\mathbb{P}, \mathbb{Q})$ is the Wasserstein distance of probability distribution $\mathbb{P}$ and $\mathbb{Q}$, $\pi(i,j)$ is the cost function and $s(i,j)$ is a distance function, $w_{q_j}$ and $w_{p_j}$ are the probabilities of $q_j$ and $p_j$ respectively.

Since we have proved $g : A \rightarrow B$ is a bijection, besides, according to our encoding rule, $g(w)$ and $w$ will corresponds to the same WEAVE, hence they will share the same representation vector. As a consequence, for each point $(g_i, w_{g_i})$ ($g_i$ corresponds to a representation vector, $w_{g_i}$ represents the probability of $g_i$) in the distribution $P_{\mathcal{G}}$, we can find a point $(h_i, w_{h_i})$ in $P_{\mathcal{H}}$ such that $g_i = h_i$, and $w_{g_i} = w_{h_i}$. Then consider (11), for $P_{\mathcal{G}}$ and $P_{\mathcal{H}}$, if we let $\pi$ be a diagonal matrix with $[w_{p_1}, w_{p_2}, \ldots, w_{p_m}]$ on the diagonal and all the other elements be 0, we can make each element in the sum $\sum_{i=1}^{m} \sum_{j=1}^{n} \pi(i,j) s(i,j)$ be 0, as this sum is supposed to be nonnegative, its minimum is 0, hence $W_1(P_{\mathcal{G}}, P_{\mathcal{H}}) = 0$, which means for two isomorphic graphs $\mathcal{G}$ and $\mathcal{H}$, their WEAVE's distributions $P_{\mathcal{G}}$ and $P_{\mathcal{H}}$ are the same.

Next we will prove the necessity of this theorem. Suppose that the Wasserstein distance between the walk representation distributions $P_{\mathcal{G}}$ and $P_{\mathcal{H}}$ is 0, we will show that graph $\mathcal{G}$ and $\mathcal{H}$ are isomorphic. Let the number of the nodes of graph $\mathcal{G}$ is $n_1$, the number of the nodes of graph $\mathcal{H}$ is $n_2$, let the number of the edges on graph $\mathcal{G}$ is $m_1$, the number if the edges on graph $\mathcal{H}$ is $m_2$. Let $k = 2\max\{m_1, m_2\} - 1$.

Now, we will give a bijective mapping $f : V(\mathcal{G}) \rightarrow v(\mathcal{H})$. First, consider the walks on graph $\mathcal{G}$, as $k = 2\max\{m_1, m_2\} - 1 \geq 2m_1 - 1$, according to Lemma 1, there exists at least one walk of length $k$ on graph $\mathcal{G}$ which can cover all the edges of $\mathcal{G}$. Consider such a walk $w_{\mathcal{G}}$, let $r_{\mathcal{G}} = [1, 2, 3, ..., t]^{\top}$ be the representation vector (corresponds to a WEAVE) we obtained according to our encoding rule. Now, we will use this representation to mark the nodes on graph $\mathcal{G}$. Mark the first node in this walk as $u_1$ (corresponds to 1 in the representation), the second node as $u_2$, the $i$th appearing node in $w_{\mathcal{G}}$ is $u_i$, continue this process untill we marked all the new appearing nodes in this walk. Since $w_{\mathcal{G}}$ can visit all the edges of graph $\mathcal{G}$, all the nodes on this graph will definitely be marked, hence the last new appearing node will be marked as $u_{n_1}$. Now, let's consider the walks on graph $\mathcal{H}$. As we have assumed that $W_1(P_{\mathcal{G}}, P_{\mathcal{H}}) = 0$, which means that for each point $(g_i, w_{g_i})$ on $P_{\mathcal{G}}$, we can find a point $(h_i, w_{h_i})$ in $P_{\mathcal{H}}$ such that $g_i = h_i$, and $w_{g_i} = w_{h_i}$. As a consequence, as $r_g$ is a point on $P_{\mathcal{G}}$, there must be a point $r_h$ on $\mathcal{H}$ such that $r_h = r_g = [1, 2, 3, ..., t]^{\top}$. Then choose any walk $w_h$ on $\mathcal{H}$ which produce $r_h$, and apply the same method to mark the nodes in this walk in order as $v_1, v_2, ..., v_{n_1}$. Now we can define the mapping $f$, let $f : V(\mathcal{G}) \rightarrow V(\mathcal{H})$, s.t., $f(u_i) = v_i$ for $\forall i \in [n_1]$, which is exactly the mapping we are looking for.

Next, we just need show for each edge $(u_i, u_j) \in E(\mathcal{G})$, we have $(f(u_i), f(u_j)) \in E(\mathcal{H})$, and vice versa, then we can prove $\mathcal{G}$ and $\mathcal{H}$ are isomorphic. The first direction is obviously true as $w_{\mathcal{G}}$ covers all the edges on $\mathcal{G}$, for any edge $(u_i, u_j)$ in $w_{\mathcal{G}}$, we have $(f(u_i), f(u_j)) = (v_i, v_j)$ which belongs to $w_h$, since $w_h$ is walk on $\mathcal{H}$, we have $(v_i, v_j) \in E(\mathcal{H})$. Then we will prove the reverse direction, i.e., for any $(v_i, v_j) = (f(u_i), f(u_j)) \in E(\mathcal{H})$, we have $(u_i, u_j) \in E(\mathcal{G})$. To prove this, we will first show that the number of edges of graph $\mathcal{G}$ and $\mathcal{H}$ are the same, i.e., $m_1 = m_2$. Suppose this is not true, without loss of generality, let $m_1 > m_2$. Since $P_{\mathcal{G}}$ and $P_{\mathcal{H}}$ are the results of random walks for infinite times. Then there must exists some walks which can visit the additional edges on $\mathcal{G}$, as a consequence, we can obtain some representation vector which will not appear

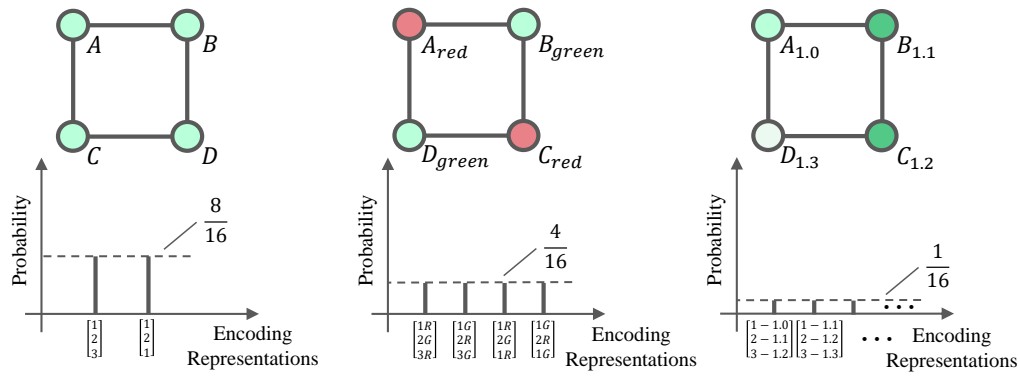

(a) Graph without node attributes      (b) Graph with discrete node attributes      (c) Graph with continuous node attributes

Figure 7: Walk representation distributions of graphs without attributes, graphs with discrete attributes, and graphs with continuous attributes.

in $P_{\mathcal{H}}$, which contradicts our assumption. Hence, we have $m_1 = m_2$. Besides, since we have $r_g = r_h$, according to Lemma 2, we can derive that the number of distinct edges on $w_g$ and $w_h$ are the same. As $w_g$ covers all the edges on $\mathcal{G}$, hence the number of distinct edges on $w_g$ is $m_1$. Therefore, the number of distinct edges on $w_h$ is also $m_1$, which means $w_h$ also has visited all the edges on $\mathcal{H}$. As for any edge $(v_i, v_j)$ on $w_h$, we have $(u_i, u_j)$ on $w_h$, in other words, we have $(u_i, u_j) = (f^{-1}(v_i), f^{-1}(v_j)) \in E(\mathcal{G})$. Hence we complete the proof. ∎

Figure 7 shows the walk representation distributions for a 4 nodes ring with walk length $k = 2$ in three different cases: without node attributes, with discrete node attributes, and with continuous node attributes. We can see the attributes will have an influence to the distributions, more specifically, the probability of each unique walk keeps the same no matter what the attributes are, however, the probability of each representation vector may vary as different unique walks may correspond to one representation vector, and the attributes may influence how many representation vectors there will be and how many unique walks correspond to a representation vector. To clarify, in Figure 7 (a), the ring graph does not have nodes attributes, there exists 16 unique walks in total, among them walk ABD, BDC, DCA, CAB, DBA, CDB, ACD, BAC will all be encoded as $r_1 = \begin{bmatrix} 1 & 2 & 3 \end{bmatrix}^{\top}$, walk ABA, BAB, BDB, DBD, CDC, DCD, CAC, ACA will be encoded as $r_2 = \begin{bmatrix} 1 & 2 & 1 \end{bmatrix}^{\top}$. Hence, for a graph in Figure 7 (a), we have $Pr(r_1) = \frac{8}{16}$, $Pr(r_2) = \frac{8}{16}$. In Figure 7 (b), each node has a discrete attribute, *i.e.*, red or green, there are still 16 unique walks in total. However, in this case, there exits four different representation vectors, walk ABC, CBA, ADC, CDA will be encoded as $r_1 = \begin{bmatrix} 1R & 2G & 3R \end{bmatrix}^{\top}$, where $R$ represents Red while $G$ represents Green; walk BCD, DCB, DAB, DCB correspond to $r_2 = \begin{bmatrix} 1G & 2R & 3G \end{bmatrix}^{\top}$; walk ABA, ADA, CDC, CBC correspond to $r_3 = \begin{bmatrix} 1R & 2G & 3R \end{bmatrix}^{\top}$; walk BAB, BCB, DCD, DAD correspond to $r_3 = \begin{bmatrix} 1R & 2G & 3R \end{bmatrix}^{\top}$. In this case, we have $Pr(r_1) = Pr(r_2) = Pr(r_3) = Pr(r_4) = \frac{4}{16}$. In the last, let's consider the case when there exists continuous nodes attributes, for such a graph, the value of nodes attributes has infinite choices, hence, it is very likely that each node may have different attribute. As a consequence, each unique walk will correspond to a unique representation vector. In our example Figure 7 (c), there also exists 16 unique walks, each walk has a particular representation vector, hence, the probability of each representation vector is $\frac{1}{16}$.

# D    PROOF FOR THEOREM 2

*Proof.* The proof for Theorem 2 is quite similar as the proof of Theorem 1, this is because the attributes just influence the representation vector form and how many unique walks correspond to a representation vector, however, the probability of each unique walk keeps same. Hence, we can use a similar method to complete the proof. Similarly, we will first prove the sufficiency. Let $\mathcal{G}$ and $\mathcal{H}$

be two isomorphic graphs with attributes, we will prove that the walk representations distribution of $\mathcal{G}$ and $\mathcal{H}$ are the same. Suppose that $A$ and $B$ are the sets of possible walks of length $k$ on $\mathcal{G}$ and $\mathcal{H}$ respectively. By applying the same analysis method as in the proof of Theorem 1, we can show that there exists a bijective mapping $g : A \to B$ such that for $\forall w_i = v_1 v_2 v_3 \ldots v_t \in A$, we have

$$g(w_i) = f(v_1)f(v_2)\ldots f(v_t) \in B, \tag{12}$$

where $f : V(\mathcal{G}) \to V(\mathcal{H})$ satisfies $\forall (v_i, v_j) \in E(\mathcal{G})$, we have $(f(v_i), f(v_j)) \in E(\mathcal{H})$ and for $\forall v_i \in V(\mathcal{G})$, the attribute of $v_i$ and $f(v_i)$ are the same. Hence, according to our encoding rule, $w_i$ and $f(w_i)$ will be encoded as the same representation vector, which means for each point $(r_{g_i}, Pr(r_{g_i}))$ in the representation distribution of $\mathcal{G}$, we can find a point $(r_{h_i}, Pr(r_{h_i}))$ in the distribution of $\mathcal{H}$ such that $r_{g_i} = r_{h_i}$, $Pr(r_{g_i}) = Pr(r_{h_i})$. Thus, we can obtain the Wasserstein distance of distribution $P_{\mathcal{G}}$ and the distribution $P_{\mathcal{H}}$ is $W_1(P_{\mathcal{G}}, P_{\mathcal{H}}) = 0$ via a similar approach as in Theorem 1. In other words, we have $P_{\mathcal{G}} = P_{\mathcal{H}}$. In addition, the necessity proof of Theorem 2 is the same as Theorem 1. ■

## E  GRAPHS WITH NODE ATTRIBUTES AND EDGE ATTRIBUTES

If both the nodes and edges in a graph have attributes, the graph is an attributed graph denoted by $\mathcal{G} = (V, E, \alpha, \beta)$, where $\alpha : V \to L_N$ and $\beta : E \to L_E$ are nodes and edges labeling functions, $L_N, L_E$ are sets of labels for nodes and edges. In this case, the graph isomorphism are defined as:

**Definition .** *Given two graphs $\mathcal{G} = (V(\mathcal{G}), E(\mathcal{G}), \alpha_{\mathcal{G}}, \beta_g)$ and $\mathcal{H} = (V(\mathcal{H}), E(\mathcal{H}), \alpha_{\mathcal{H}}, \beta_{\mathcal{H}})$, then $\mathcal{G}$ and $\mathcal{H}$ are isomorphic with node attributes as well as edge attributes if there is a bijection $f : V(\mathcal{G}) \Leftrightarrow V(\mathcal{H})$*

$$\forall uv \in E(\mathcal{G}) \Leftrightarrow f(u)f(v) \in E(\mathcal{H}), \tag{13}$$
$$\alpha_{\mathcal{G}}(u) = \alpha_{\mathcal{H}}(f(u)), \forall u \in V(\mathcal{G}), \tag{14}$$
$$\beta_{\mathcal{G}}(u,v) = \beta_{\mathcal{H}}(f(u), f(v)). \tag{15}$$

**Corollary 1.** *Let $\mathcal{G} = (V(\mathcal{G}), E(\mathcal{G}))$ and $\mathcal{H} = (V(\mathcal{H}), E(\mathcal{H}))$ be two connected graphs with node attributes. Suppose we can enumerate all possible WEAVEs on $\mathcal{G}$ and $\mathcal{H}$ with a fixed-length $k \geq 2\max\{|E(\mathcal{G})|, |E(\mathcal{H})|\} - 1$, where each WEAVE has a unique vector representation generated from a well-trained autoencoder. The Wasserstein distance between $\mathcal{G}$'s and $\mathcal{H}$'s WEAVE distributions is $0$ if and only if $\mathcal{G}$ and $\mathcal{H}$ are isomorphic with both node attributes and edge attributes.*

*Proof.* When both nodes and edges of a graph are given attributes, the representation vectors of random walks will be different. However, just like the cases with only nodes attributes, the probability of each unique walk on the graph keeps same. Hence, we can follow a similar analysis method as Theorem 2 to complete this proof. ■

| Dataset | Identity Kernel | | | DeepSet-MMD | | |
| | Classification | Clustering | | Classification | Clustering | |
| | ACC | ACC | NMI | ACC | ACC | NMI |
| --- | --- | --- | --- | --- | --- | --- |
| NCI1 | 0.6105 | 0.5510 | 0.0073 | 0.6382 | 0.5630 | 0.0095 |
| PROTEINS | 0.7207 | 0.5957 | 0.0518 | 0.7103 | 0.5965 | 0.0438 |
| COLLAB | 0.6720 | 0.5973 | 0.2108 | 0.6572 | 0.5668 | 0.2015 |
| IMDB-BINARY | 0.7660 | 0.5776 | 0.0241 | 0.7210 | 0.5219 | 0.0225 |
| IMDB-MULTI | 0.4466 | 0.3816 | 0.0214 | 0.4258 | 0.3647 | 0.0168 |

Table 5: Representation evaluation based on classification and clustering down-stream tasks

## F  DEEPSET IN THE COMPONENT OF EMBEDDING DISTRIBUTIONS

In this section, we investigate whether DeepSet (Zaheer et al., 2017) is an effective technique for distribution embedding. In particular, we employ DeepSet to replace the multi-layer neural network for feature mapping function approximation, and similarity values generated by MMD serve

as supervision signals to guide DeepSet training. In our experiments, we compare the SEED implementation based on DeepSet with MMD (DeepSet in Table 5) with the SEED implementation based on the identity kernel (Identity Kernel in Table 5). We also observe that the MMD does not have significant performance difference. The result confirms that DeepSet could be a strong candidate for the component of Embedding subgraph distributions.

## G ABLATION STUDY ON WEAVE

| Feature utilized | Classification ACC | Clustering ACC | Clustering NMI |
|---|---|---|---|
| Only node feature | 0.6444 | 0.6744 | 0.0625 |
| Only earliest visit time | 0.8112 | **0.8014** | **0.3214** |
| Node feature + Earliest visit time | **0.8222** | 0.7260 | 0.1567 |

Table 6: The impact of node feature and earliest visit time in WEAVE based on MUTAG dataset

In this section, we investigate the impact of node features and earliest visit time in WEAVE. In Table 6, *Only node feature* means only node features in WEAVE are utilized for subgraph encoding (which is equivalent to vanilla random walks), *only earliest visit time* means only earliest visit time information in WEAVE is used for subgraph encoding, and *Node feature + earliest visit time* means both information is employed. We evaluate the impact on the MUTAG dataset. As shown above, it is crucial to use both node feature and earliest visit time information in order to achieve the best performance. Interestingly, on the MUTAG dataset, we observe that clustering could be easier if we only consider earliest visit time information. On the MUTAG dataset, node features seem to be noisy for the clustering task. As the clustering task is unsupervised, noisy node features could negatively impact its performance when both node features and earliest visit time information are considered.

## H NYSTRÖM APPROXIMATION IN THE SEED FRAMEWORK

In this section, we evaluate the impact of Nyström based kernel approximation (Williams & Seeger, 2001) to the component of embedding distributions.

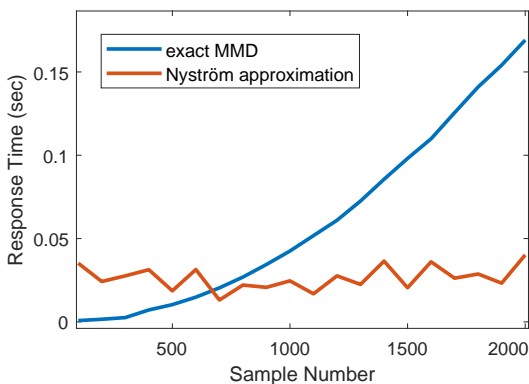

Figure 8: Response time comparison between exact MMD and its Nyström approximation

First, we investigate the impact to the effectiveness in the downstream tasks. In this set of experiment, we implement a baseline named SEED-Nyström, where the Nyström method is applied to approximate RBF kernel based MMD during training phases with 200 sampled WEAVEs. In particular, top 30 eigenvalues and the corresponding eigenvectors are selected for the approximation. As shown in Table 7, across five datasets, SEED-Nyström achieves comparable performance, compared with the case where an identity kernel is adopted.

In addition, we evaluate the response time of exact RBF kernel based MMD and its Nyström approximation. Top 30 eigenvalues and the corresponding eigenvectors are selected for the Nyström

| Dataset | RBF Kernel | | | SEED-Nyström | | |
|---------|----------------|----------|------|----------------|----------|------|
| | Classification | Clustering | | Classification | Clustering | |
| | ACC | ACC | NMI | ACC | ACC | NMI |
| NCI1 | 0.6211 | 0.5610 | 0.0079 | 0.6281 | 0.5518 | 0.0081 |
| PROTEINS | 0.7161 | 0.5857 | 0.0476 | 0.7054 | 0.5738 | 0.0389 |
| COLLAB | 0.6718 | 0.5212 | 0.1831 | 0.6447 | 0.5217 | 0.1983 |
| IMDB-BINARY | 0.7421 | 0.5582 | 0.0218 | 0.7280 | 0.5018 | 0.0211 |
| IMDB-MULTI | 0.4541 | 0.3985 | 0.0241 | 0.4148 | 0.3676 | 0.0172 |

Table 7: Representation evaluation based on classification and clustering down-stream tasks

approximation. As shown in Figure 8, when we range the number of WEAVE samples from 100 to 2000, the Nyström approximation scales better than the exact MMD evaluation.

In summary, the Nyström method is a promising method that can further improve the scalability of the SEED framework in training phases, especially for the cases where a large number of WEAVE samples are required.

