# OpenReview forum: "Inductive and Unsupervised Representation Learning on Graph Structured Objects"
_ICLR.cc/2020/Conference — Accept (Poster)_

### Official Review · AnonReviewer1 · 2019-10-18
**Official Blind Review #1**

**Rating:** 6

**Review:**

The authors propose a method for learning graph embeddings and focus specifically on a setting where not all graphs are part of the training data (the inductive setting). The core problem of graph embedding methods is to find a learnable function that maps arbitrary graphs into a fixed-sized vector representation. There have been several proposals ranging from the class of graph kernels to variations of graph neural networks. The authors propose a method that consists of three steps

(1) sample a number of subgraphs from the original graphs
(2) learn an encoding function for these subgraphs (subgraph -> vector representation)
(3) for every graph we, therefore, get a set of vector representations, one per subgraph. We now try to find a similarity measure operating on sets of vectors to compute the distance between graphs.

The novel bits are
(a) the way that the subgraphs are sampled (using an algorithm called WEAVE, that stores more information about random walks) and, therefore, is able to be distinguish graphs based on the extracted walks that standard random walk based methods cannot; and
(b) to define a similarity measure based on the set of vectors.
The authors also prove that their method is (under some assumptions) able to decide the isomorphism problem. This is a nice result to have in light of recent papers that have investigated the limitations of GNN in comparison to Weisfeiler-Leman and isomoprhism testing.

Unfortunately, the proposed method has limited novelty. The WEAVE sampling is a small variation on random walk sampling that's been around for a while in graph representation learning. Also, to define the similarity between set of vectors has been addressed before in numerous papers (e.g., all papers investigating learning for sets, DeepSets, etc.) and the method here seems a bit ad-hoc and doesn't compare to existing work.

My "novelty" critique is also made in light of the small number of datasets on which experiments have been conducted. If a new simple random walk strategy would lead to clearly better results on a number of datasets, this would be a significant contribution. As far as I can tell, however, the results are mixed and not very impressive especially due to the small number of datasets.

**Experience Assessment:**

I have published in this field for several years.

**Review Assessment: Checking Correctness Of Derivations And Theory:**

I assessed the sensibility of the derivations and theory.

**Review Assessment: Checking Correctness Of Experiments:**

I assessed the sensibility of the experiments.

**Review Assessment: Thoroughness In Paper Reading:**

I read the paper thoroughly.

---

> ### Author Response · Authors · 2019-11-13
> **Response to the comments from the reviewer (Part 1)**
>
> We sincerely appreciate your valuable comments to our work.
>
> Q1. Unfortunately, the proposed method has limited novelty. The WEAVE sampling is a small variation on random walk sampling that's been around for a while in graph representation learning. Also, to define the similarity between set of vectors has been addressed before in numerous papers (e.g., all papers investigating learning for sets, DeepSets, etc.) and the method here seems a bit ad-hoc and doesn't compare to existing work.
>
> Thanks for sharing your concerns. There could be confusion on the contribution in our work. Therefore, we discuss this concern from the following aspects.
>
> $\bullet$ $\small\textbf{The main technical contribution.}$ We address the problem of inductive and unsupervised graph representation learning. As it is intractable to evaluate the error between input and reconstructed graphs, it is challenging to make graph learning inductive and unsupervised simultaneously. We propose the framework SEED, and its core idea is novel.
>     1. Instead of directly evaluating reconstruction errors for original graphs, we first sample subgraphs which can preserve structural information and lead to efficient reconstruction evaluation.
>     2. With the observation that similar graphs share similar subgraph distribution, we use the embedding of an input graph's subgraph distribution as its vector representation.
> For concrete implementations, we need to address three questions
>     I. What is the subgraph?
>     II. How to encode such a subgraph?
>     III. How to embed subgraph distributions?
> In this work, we propose a competitive implementation with three concrete components to answer the questions.
>
> $\bullet$ $\small \textbf{What is the novelty in WEAVE?}$ WEAVE is our answer to Question I. WEAVE is a random walk variant that has the capability to preserve loop information in traversed graph data. As discussed in the paper, when the SEED framework is geared with WEAVE, it becomes closely related to graph isomorphism; however, existing random walk variants cannot meet this goal. While existing random walk variants have been widely utilized in node representation learning (e.g., DeepWalk and so on), WEAVE is the one that enables inductive and unsupervised graph representation learning. Note that our goal is not to propose a new random walk variant. Instead, our goal is to propose a strong candidate that meets the requirements in SEED.
>
> $\bullet$ $\small \textbf{Are existing set similarity techniques related?}$ Existing set similarity techniques, such as DeepSet, could be quite related. In particular, DeepSet could be another strong candidate for the step of embedding subgraph distributions. In the latest draft, we have added a new baseline named DeepSet, where DeepSet is adopted for embedding subgraph distributions in SEED. The detailed empirical study is presented in Appendix I. We briefly summarize our discovery as follows.
>
> $\bullet$ The SEED implementation based on DeepSet achieves competitive performance compared with the implementation based on identity kernel. The results suggest the effectiveness of DeepSet in the SEED framework.

---

> > ### Comment · AnonReviewer1 · 2019-11-15
> > **Response to the rebuttal**
> >
> > Dear authors,
> >
> > Thank you for providing a rebuttal and adding new experimental results. I think that this has strengthened the paper and I increase my score to a weak accept. The novelty of the proposed approach is still limited but now with more empirical results and comparisons to existing methods, it is less of a concern.

---

> ### Author Response · Authors · 2019-11-13
> **Response to the comments from the reviewer (Part 2)**
>
> Q2. My "novelty" critique is also made in light of the small number of datasets on which experiments have been conducted. If a new simple random walk strategy would lead to clearly better results on a number of datasets, this would be a significant contribution. As far as I can tell, however, the results are mixed and not very impressive especially due to the small number of datasets.
>
> Thanks for your suggestions. In the latest draft, we have strengthened the experimental study in this paper, including the following updates.
> $\bullet$ We have added five additional public benchmark datasets, including NCI1, PROTEINS, COLLAB, IMDB-BINARY, and IMDB-MULTI. The dataset description is updated in Appendix F, and the their evaluation results are presented in Appendix G.
> $\bullet$ We have added another set of ablation study, where we evaluate the impact of different features in WEAVE. The evaluation results are presented in Appendix J.
> $\bullet$ We have added a new baseline where DeepSet serves for embedding subgraph distribution in SEED. In Appendix I, we investigate whether DeepSet is also a good option in the step of embedding distribution.
>
> The newly added experimental results are detailed in Appendix F, G, I, and J. In the following, we briefly summarize the key observations.
> $\bullet$ The SEED framework outperforms the baseline methods in 18 out of 21 cases, and achieves competitive performance in the rest 3 cases. In particular, the SEED achieves up to 0.18, 0.13, and 0.22 absolute performance improvement, in terms of classification accuracy, clustering accuracy, and clustering NMI, respectively.
> $\bullet$ WEAVEs consistently outperforms vanilla random walks (without earliest visit time information). Indeed, WEAVEs are stronger at preserving structural information, while loops information is usually lost in vanilla random walks. The results highlight the importance of WEAVEs in the SEED framework.
> $\bullet$ For the baseline where DeepSet is deployed in the component of embedding subgraph distributions, we observe that it achieves similar performance compared with the one using feature mapping function evaluation. We confirm that DeepSet is compatible with the SEED framework, and it could be a good candidate in the step of embedding subgraph distributions.

---

### Official Review · AnonReviewer3 · 2019-10-22
**Official Blind Review #3**

**Rating:** 6

**Review:**

The authors propose a general framework SEED (Sampling, Encoding, and Embedding Distributions) for inductive and unsupervised representation learning on graph structured objects. The most innovative part in this paper is random walks with earliest visiting time (WEAVE) in the sampling part. WEAVE has potential power for capturing structure difference and could reflect isomorphism as well. Instead of using language model like word2vec, encoding part leverages MLP to get the embedding of each WEAVE, which is efficient and intuitive. Then, the group of encoding results from all WEAVEs are aggregated with kernel functions, generating the final embedding of a graph. This method achieves better accuracy in both clustering and classification tasks than previous ones including GraphSAGE, GMN and GIN.
This method uses an elegant way to embed graphs in an unsupervised manner, and the new random walk approach provides insights into graph structure encoding. Factors like walk length and number of walks are strictly derived then well examined in real experiments.
Questions & suggestions:
1 In WEAVE encoding part, the paper doesn’t show how much the earliest visiting time information improve the model. In another word, if we leave out the timing term (x_t^{(p)}), will the model still perform well?
2 In embedding distribution part, it seems that only identity kernel is easy to calculate. For commonly adopted kernels, the MLP is “mimicking the behavior of a kernel”, so it will still be limited by the kernel you choose in MMD. There are some statistical approaches to estimate Ф(z) like Nystrom method, maybe that will be another solution other than taking average.

**Experience Assessment:**

I have read many papers in this area.

**Review Assessment: Checking Correctness Of Derivations And Theory:**

I assessed the sensibility of the derivations and theory.

**Review Assessment: Checking Correctness Of Experiments:**

I assessed the sensibility of the experiments.

**Review Assessment: Thoroughness In Paper Reading:**

I read the paper at least twice and used my best judgement in assessing the paper.

---

> ### Author Response · Authors · 2019-11-13
> **Response to the comments from the reviewer (Part 1)**
>
> Thank you so much for recognizing our work. We sincerely appreciate your valuable comments. The following are our response to your questions or concerns.
>
> Q1. In WEAVE encoding part, the paper doesn't show how much the earliest visiting time information improve the model. In another word, if we leave out the timing term $x_t^{(p)}$, will the model still perform well?
>
> The earliest visiting time information is critical for WEAVE. Compared with vanilla random walk, WEAVE is able to preserve loop information in its traversed graph data because of the earliest visiting time.
>
> In the latest draft, we have added Appendix J. In particular, we have added a new baseline where WEAVEs without the earliest visiting time information are employed for subgraph sampling and encoding. We briefly summarize our observations as follows.
> $\bullet$ The classification and clustering performance could suffer significant performance drop if we only consider node features for subgraph encoding.
> $\bullet$ We achieve the best performance when we jointly consider both node feature and earliest visit time information.
>
> In addition, we have strengthened the experimental study in this paper, including the following updates.
> $\bullet$ We have added five additional public benchmark datasets, including NCI1, PROTEINS, COLLAB, IMDB-BINARY, and IMDB-MULTI. The dataset description is updated in Appendix F, and the evaluation results are presented in Appendix G.
> $\bullet$ We have added another set of ablation study, where we evaluate the impact of different features in WEAVE. The evaluation results are presented in Appendix J.
> $\bullet$ We have added a new baseline where DeepSet serves for embedding subgraph distribution in SEED. In Appendix I, we investigate whether DeepSet is a good option in the step of embedding distribution.
>
> The newly added experimental results are detailed in Appendix F, G, I, and J. In the following, we briefly summarize the key observations.
> $\bullet$ The SEED framework outperforms the baseline methods in 18 out of 21 cases, and achieves competitive performance in the rest 3 cases. In particular, the SEED achieves up to 0.18, 0.13, and 0.22 absolute performance improvement, in terms of classification accuracy, clustering accuracy, and clustering NMI, respectively.
> $\bullet$ WEAVEs consistently outperforms vanilla random walks (without earliest visit time information). Indeed, WEAVEs are stronger at preserving structural information, while loops information is usually lost in vanilla random walks. The results highlight the importance of WEAVEs in the SEED framework.
> $\bullet$ For the baseline where DeepSet is deployed in the component of embedding subgraph distributions, we observe that it achieves similar performance compared with the one using feature mapping function evaluation. We confirm that DeepSet is compatible with the SEED framework, and it could be a good candidate in the step of embedding subgraph distributions.
>
> Q2. In embedding distribution part, it seems that only identity kernel is easy to calculate. For commonly adopted kernels, the MLP is “mimicking the behavior of a kernel”, so it will still be limited by the kernel you choose in MMD. There are some statistical approaches to estimate Ф(z) like Nyström method, maybe that will be another solution other than taking average.
>
> Thanks for this good suggestion. It could be promising to employ Nyström method in the SEED framework. We will provide a concrete discussion to this question in part 2 of our response.

---

> ### Author Response · Authors · 2019-11-14
> **Response to the comments from the reviewer (Part 2)**
>
> Q2. In embedding distribution part, it seems that only identity kernel is easy to calculate. For commonly adopted kernels, the MLP is “mimicking the behavior of a kernel”, so it will still be limited by the kernel you choose in MMD. There are some statistical approaches to estimate Ф(z) like Nyström method, maybe that will be another solution other than taking average.
>
> Thanks for this good suggestion. In the latest draft, we have added Appendix K, where we evaluate how the Nyström approximation impacts the effectiveness, and its scalability advantage. From the effectiveness and efficiency results, we see that the Nyström method is promising to further enhance the scalability of the SEED framework in training phases, especially for the cases where a large number of WEAVE samples are needed.

---

### Official Review · AnonReviewer2 · 2019-10-25
**Official Blind Review #2**

**Rating:** 6

**Review:**

In this work, a novel graph similarity learning framework SEED is proposed. Given an input graph, SEED proceeds in 4 steps, namely Sampling subgraphs, Encoding sampled subgraphs using autoencoder, aggregating subgraphs' Embedding Distribution into a vector representation.  Theretically, a connection between proposed SEED and the graph isomorphism is established. Experimentally, simulation on DEEZE and MUTAG datasets validated the effectivety of the proposed graph learning framework.

Pro:
The paper is well structured and easy to follow.  Experiments appears convincing, especially the t-SNE plots when varying the number of subgraph samples.

Con:
It would be better if experiments can be conducted on a few more benchmark datasets used in the compared methods.

Minor:
Last sentence in Page 6: each component have been -> each component has been
the 5-th sentence in Sec. 4.3, focusing -> focuses
the 7-th sentence in Sec. 4.3, At the meantime -> In the meantime

**Experience Assessment:**

I have read many papers in this area.

**Review Assessment: Checking Correctness Of Derivations And Theory:**

I did not assess the derivations or theory.

**Review Assessment: Checking Correctness Of Experiments:**

I carefully checked the experiments.

**Review Assessment: Thoroughness In Paper Reading:**

I read the paper at least twice and used my best judgement in assessing the paper.

---

> ### Author Response · Authors · 2019-11-13
> **Response to the comments from the reviewer**
>
> Thank you so much for recognizing our work. We sincerely appreciate your valuable comments. Our answers to your questions/concerns are as follows.
>
> Q1: It would be better if experiments can be conducted on a few more benchmark datasets used in the compared methods.
>
> Thanks for the good suggestion. In the latest draft, we have strengthened the experimental study in this paper, including the following updates.
> $\bullet$ We have added five additional public benchmark datasets, including NCI1, PROTEINS, COLLAB, IMDB-BINARY, and IMDB-MULTI. The dataset description is updated in Appendix F, and the  evaluation results are presented in Appendix G.
> $\bullet$ We have added another set of ablation study, where we evaluate the impact of different features in WEAVE. The evaluation results are presented in Appendix J.
> $\bullet$ We have added a new baseline where DeepSet serves for embedding subgraph distribution in SEED. In Appendix I, we investigate whether DeepSet is a good option in the step of embedding distribution.
>
> The newly added experimental results are detailed in Appendix F, G, I, and J. In the following, we briefly summarize the key observations.
> $\bullet$ The SEED framework outperforms the baseline methods in 18 out of 21 cases, and achieves competitive performance in the rest 3 cases. In particular, the SEED achieves up to 0.18, 0.13, and 0.22 absolute performance improvement, in terms of classification accuracy, clustering accuracy, and clustering NMI, respectively.
> $\bullet$ WEAVEs consistently outperforms vanilla random walks (without earliest visit time information). Indeed, WEAVEs are stronger at preserving structural information, while loops information is usually lost in vanilla random walks. These results highlight the importance of WEAVEs in the SEED framework.
> $\bullet$ For the baseline where DeepSet is deployed in the component of embedding subgraph distributions, we observe that it achieves similar performance compared with the one using feature mapping function evaluation. We confirm that DeepSet is compatible with the SEED framework, and it could be a good candidate in the step of embedding subgraph distributions.
>
> In addition, we have done another round of proof reading, and have fixed typos or grammar errors, including those suggested by the reviewers.

---

### Author Response · Authors · 2019-11-15
**Appreciate your attention and time**

Dear Reviewers,

We sincerely appreciate your valuable comments that help us improve the paper. If you have more questions or concerns about our latest draft or response, please feel free to let us know. We are happy to discuss with you.

---

### Decision · Program_Chairs · 2019-12-19

**Decision:**

Accept (Poster)

**Comment:**

The paper focuses on the problem of finding dense representations of graph-structured objects in an unsupervised manner. The authors propose a novel framework for solving this problem and show that it improves over competitive baselines. The reviewers generally liked the paper, although were concerned with the strength of the experimental results. During the discussion phase, the authors bolstered the experimental results. The reviewers are satisfied with the resulting paper and agree that it should be accepted.